SOFTWARE

# Quantifying the clusterness and trajectoriness of single-cell RNA-seq data

**Hong Seo Lim, Peng Qiu**⊙ *

Wallace H. Coulter Department of Biomedical Engineering, Georgia Institute of Technology and Emory University, Atlanta, Georgia, United States of America

* peng.qiu@bme.gatech.edu

## Abstract

Among existing computational algorithms for single-cell RNA-seq analysis, clustering and trajectory inference are two major types of analysis that are routinely applied. For a given dataset, clustering and trajectory inference can generate vastly different visualizations that lead to very different interpretations of the data. To address this issue, we propose multiple scores to quantify the "clusterness" and "trajectoriness" of single-cell RNA-seq data, in other words, whether the data looks like a collection of distinct clusters or a continuum of progression trajectory. The scores we introduce are based on pairwise distance distribution, persistent homology, vector magnitude, Ripley's K, and degrees of connectivity. Using simulated datasets, we demonstrate that the proposed scores are able to effectively differentiate between cluster-like data and trajectory-like data. Using real single-cell RNA-seq datasets, we demonstrate the scores can serve as indicators of whether clustering analysis or trajectory inference is a more appropriate choice for biological interpretation of the data.

**Data Availability Statement:** The code supporting the findings of this study is available at https://github.com/pqiu/Quantifying_clusterness_trajectoriness.

## Author summary

Single-cell sequencing technologies have motivated development of numerous computational algorithms. Two main types of these algorithms are clustering and trajectory inference. When scientists have a scRNA-seq dataset, they usually pick one of these approaches based on what they think the data shows. If they think the data has distinct clusters of cells, they will analyze the data using clustering algorithms. If they think the data shows a continuous progression, they will use trajectory inference algorithms. However, sometimes using clustering and trajectory inference on the same data can lead to very different interpretations, where clustering algorithms produce distinct cell clusters while trajectory inference on the same data show continuous trajectories. This makes us wonder: which way of looking at the data is more appropriate? In this paper, we developed a pipeline for quantifying the "clusterness" and "trajectoriness" of scRNA-seq data, in other words, whether the data looks like a collection of distinct clusters or a continuum of progression trajectory. We think such geometric quantification is an important question that should be broadly discussed in the single-cell research community.

**Funding:** This publication is part of the Gut Cell Atlas Crohn's Disease Consortium funded by The Leona M. and Harry B. Helmsley Charitable Trust and is supported by a grant from Helmsley to Georgia Institute of Technology (www. helmsleytrust.org/gut-cell-atlas/). This work was also supported by the National Science Foundation (CCF2007029). The funders had no role in study design, data collection and analysis, the decision to publish, or the preparation of the manuscript.

**Competing interests:** The authors have declared that no competing interests exist.

## Introduction

Advances in sequencing technologies have led to new opportunities that could provide valuable insights into diverse biological systems [1], especially single-cell RNA sequencing (scRNA-seq) which enables biologists to analyze gene expression measurement at the single-cell level. Such single-cell measurements provide high resolution to reveal the cellular heterogeneity in various biological tissues and systems, which was not possible in the traditional bulk sequencing [2]. A great number of scRNA-seq technologies [3–7], such as SMART-seq, CEL-seq, Dropseq, were developed in recent years, and have been applied to investigate diverse biological systems including cellular identification, immunophenotyping, regulatory mechanism, and development trajectories [8–11].

With the rapidly growing usage of scRNA-seq technologies, various computational analysis tools have been developed. Although there exist numerous computational questions in the context of scRNA-seq analysis, many of the existing algorithms could be categorized into two major types: algorithms developed for clustering analysis [12] and algorithms developed for trajectory inference [13]. Clustering tools are primarily used to identify distinct cell types in heterogeneous cell populations. The key assumption of clustering-based algorithms is that the data contains clusters of cells where each cluster exhibits some unique gene expression pattern that is different from the other clusters. Hence, focusing on the differences among the clusters, most of the clustering algorithms try to minimize intra-cluster differences while maximizing inter-cluster differences. A few examples of clustering algorithms include community detection in Seurat [3], Co-occurrence clustering [14], and pcaReduce [15]. In contrast, trajectory inference algorithms are primarily used for extracting continuum of changes that represent dynamics of biological processes, such as cell cycle, cell differentiation, disease progression, and drug response. The main assumption of trajectory inference is that the data contains cells that represent the transitions among well-defined cellular states, and the transitions are manifested as gradual changes in the gene expression space. Therefore, trajectory inference algorithms try to capture the continuous global structure by focusing on the local connections, hence, ordering cells along a continuous path resembling the evolution of the cellular process. PAGA [16], Monocle [11], Slingshot [17], DPT [18] are a few examples of trajectory inference methods [19].

As the two different types of algorithms seek different characteristics based on their respective underlying assumptions, clustering and trajectory inference algorithms may generate vastly different visualizations and interpretations. Figs 1 and 2 illustrate eight example datasets analyzed by both types of algorithms. The first two panels in each row show visualizations from clustering analysis based on the Louvain algorithm implemented in Seurat, and from trajectory inference using the diffusion pseudo time (DPT). In the four example datasets shown in Fig 1A–1D, clustering and trajectory inference produced relatively similar visualizations and interpretations of the geometry of the data, where both types of analysis suggested that the first two datasets are trajectory-like and the remaining two datasets are cluster-like. In contrast, the four example datasets in Fig 2A–2D showed the opposite, where clustering analysis identified clearly distinct cell clusters, while trajectory inference showed continuous progression of cells, meaning that the two types of analysis led to different and potentially misleading interpretations of the same data. Therefore, the proper choice of analysis type and interpretation is a critical consideration for scRNA-seq, depending on the geometric characteristics of the data. However, there have not been any objective or data-driven methods for informing biologists whether clustering or trajectory inference is more appropriate for interpreting the data. Although there exist statistical methods to test the existence or prominence of cluster structure (e.g., such as Hopkins statistics [20], gap statistics [21], silhouette coefficient [22], Calinski-

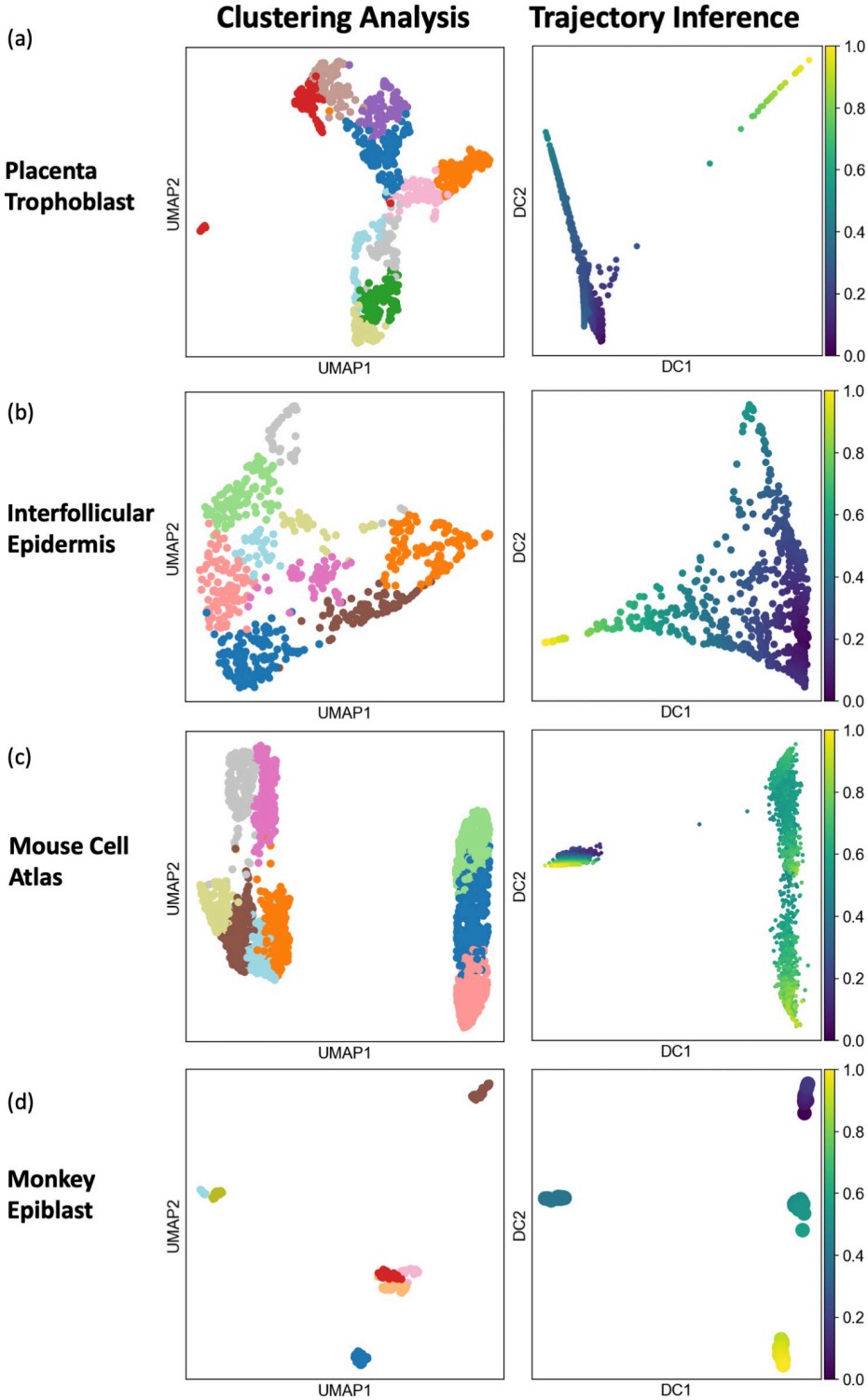

**Fig 1. Visualizations of datasets where clustering and trajectory inference produce similar interpretations.** (a, b) Two datasets with clear trajectory-like visualizations from both clustering and trajectory inference. (c, d) Two datasets with clear clustering-like visualizations from both clustering and trajectory inference.

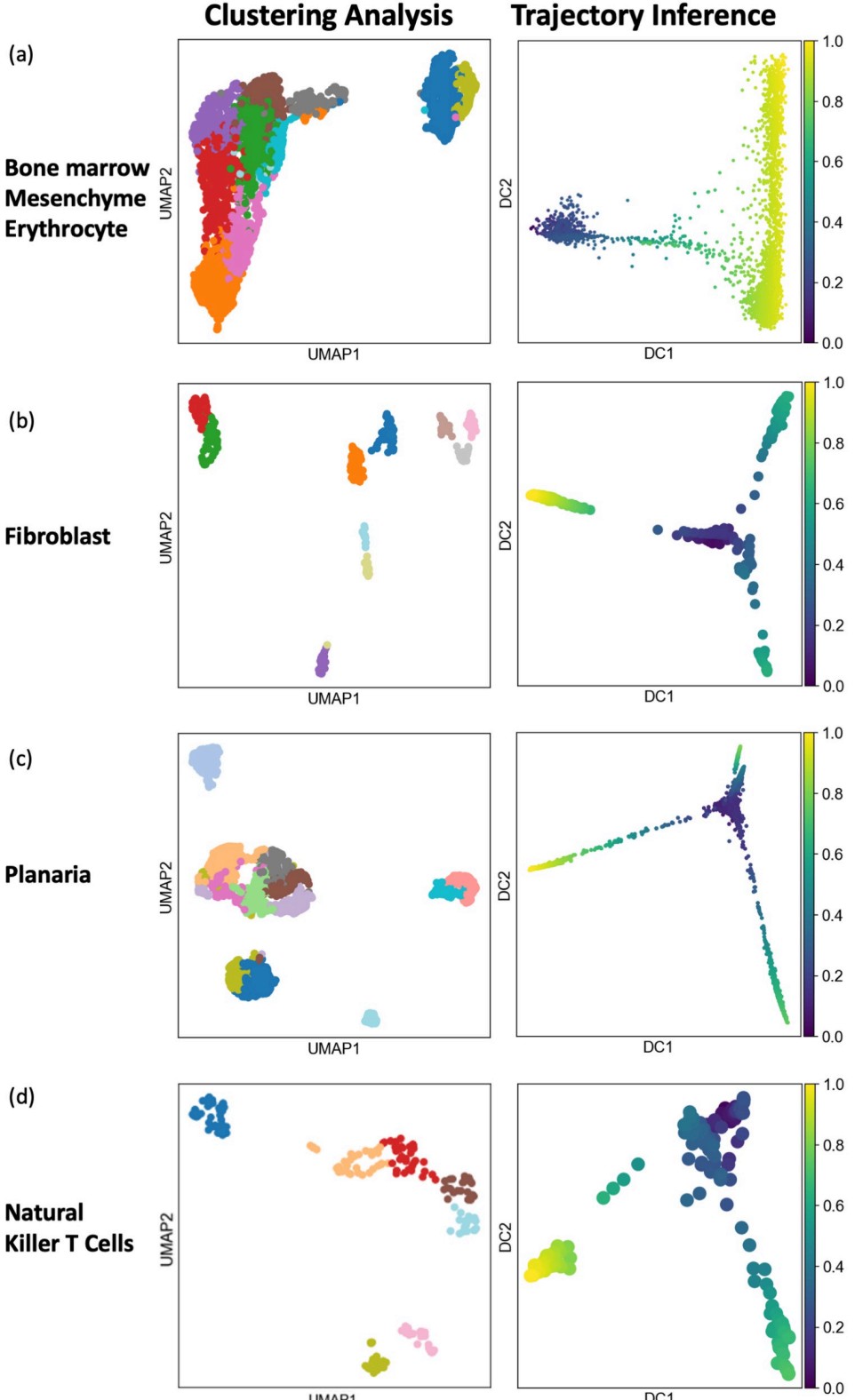

**Fig 2. Visualizations of datasets where clustering and trajectory inference produce different interpretations.** When applied to these four datasets, clustering analysis showed distinctive clusters while trajectory inference showed uninterrupted continuum, as shown in the two panels in all four cases.

Harabasz Index [23]), those methods were not designed to distinguish cluster-like and trajectory-like patterns.

Here, we present several scoring metrics that aim to quantify the "clusterness" and "trajectoriness" of the data, in other words, whether the data looks like a collection of distinct clusters or a continuum of gradual changes. The metrics were designed based on the distribution of cell-to-cell distances, persistent homology [24], vector magnitude directionality, Ripley's K function [25], and degrees of connectivity. Along with the scoring metrics, we present a pipeline that can evaluate scRNA-seq datasets in terms of whether their geometry is cluster-like or trajectory-like. Using simulated datasets, we demonstrate that the proposed metrics could be used to differentiate between the clustering-like datasets and trajectory-like datasets. Using real scRNA-seq datasets, we show that the metrics could serve as indicators of whether clustering or trajectory inference is appropriate for interpreting the geometry of those datasets.

## Design and implementation

We propose five scoring metrics to quantify the "clusterness" and "trajectoriness" of scRNA-seq data. An overview is shown in Fig 3, with each scoring metric providing a single scalar score that aims to capture a geometric characteristic of the data.

## Scoring metric 1 –distribution of pairwise distances

The first scoring metric is the entropy of the distribution of pairwise distances among the data points. This score was devised based on the following intuition. In a dataset that contains distinct clusters, if we compute the pairwise distances between all possible pairs of data points, the distribution of the distances should be bi-modal or multi-modal, because the distances are either short (between the data points within the same cluster) or long (between data points in different clusters). In contrast, in a dataset that represents a continuous trajectory, the distribution of pairwise distances is more spread out and smooth, without large fluctuations that lead

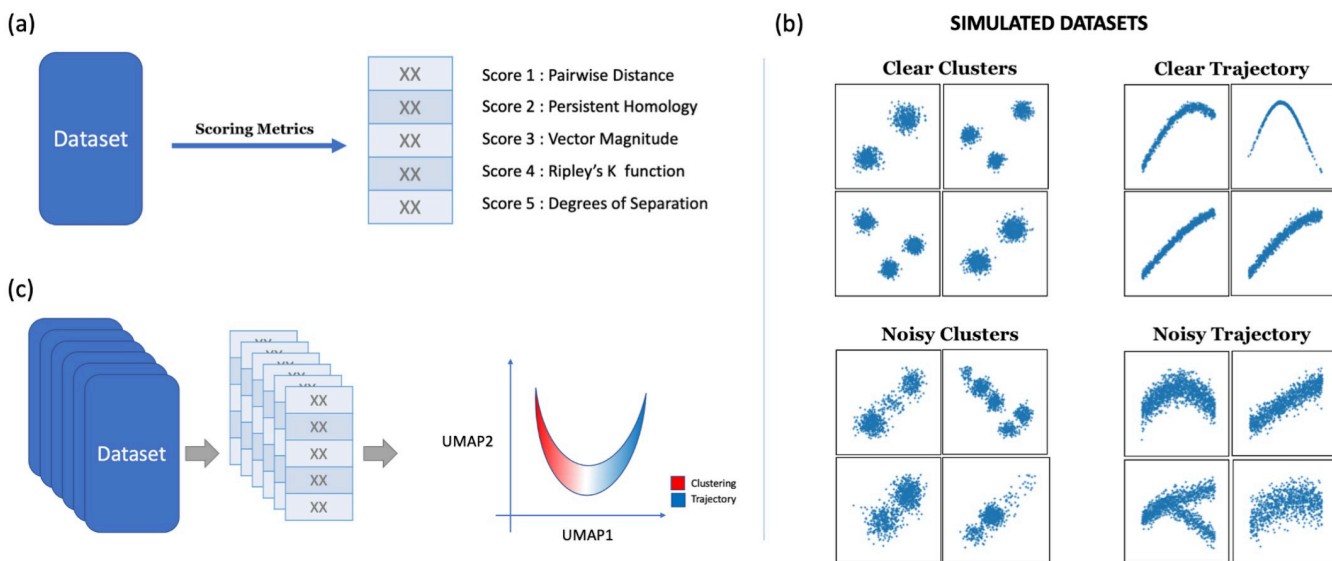

**Fig 3. Overview of the proposed pipeline and simulated data used.** (a) Given a dataset, five different scoring metrics are used to quantify the dataset. The output is five numerical scores. (b) Scatter plots visualizing a few examples of simulated datasets. The simulated datasets are two-dimensional. (c) A multitude of simulated datasets was scored by the scoring metrics, and the scores are projected to UMAP space.

to clear bi-modality or multi-modality. Therefore, the distribution of pairwise distances in cluster-like data should exhibit larger fluctuation and thus lower entropy, compared to trajectory-like data.

Since scRNA-seq data are high-dimensional and the underlying geometry is likely nonlinear, Euclidean distance may not be able to sufficiently capture the differences between cluster-like and trajectory-like datasets. Instead, we decided to use geodesic distances known to be less affected by dimensionality and nonlinearity. In particular, we used the dpt distance from the Diffusion Pseudotime (DPT) analysis, which is a scRNA-seq trajectory inference algorithm. We used the *Scanpy* [26] python package to generate the pairwise dpt distance matrix. dpt distances with infinity values are substituted with 1.5 times the maximum distances calculated. Once all the pairwise distances were calculated, we created a histogram of the distances. Our default bin size of the histogram is 10. The histogram is then normalized so that the sum of all bins adds up to 1. The entropy of the normalized histogram is our scoring metric. In general, the entropy scores for trajectory-like datasets are expected to be larger than the entropy scores for cluster-like datasets.

### Scoring metric 2 –persistent homology

The second scoring metric is derived from persistent homology. Persistent homology is a concept in topological data analysis to study qualitative features of data that persist across multiple scales [27]. In this study, we specifically used 0-dimensional persistent homology which focuses on the 'birth' and 'death' of connected components as a function of increasing the distance threshold around each data point in point clouds. To compute the 0-dimensional persistent homology, an increasing distance threshold is applied to the data. At each distance threshold value, cells are connected if their distances are smaller than the threshold. At threshold 0, all cells form disjoint components. As the threshold increases, the disjoint components merge and eventually form one connected graph. Thresholds at which merges occur are recorded, and the entropy of the distribution of the recorded thresholds serves as our scoring metric. In a cluster-like dataset, data points and components within the same clusters quickly merge at small threshold values, while components representing different clusters merge at substantially larger threshold values due to the gaps between distinct cell clusters. In contrast, in a trajectory-like dataset, the majority of the merges occur at relatively small threshold values. Therefore, the distribution of thresholds for the merges should be more spread out in cluster-like data and should be more uniformly distributed for trajectory-like data.

We used the *Ripser* [28] python package for calculating the persistent homology and used the geodesic distance matrix from DPT generated from the *Scanpy* package as distance measures, rather than the Euclidean or Manhattan distance that are often used in persistent homology. Given a dataset, the *Ripser* package outputs a vector with each element representing at what distance threshold merging of components occurred. As in the first scoring metric, we created a histogram of the vector, normalized the histogram to sum up to 1, and calculated its entropy. The natural log of the calculated entropy serves as our scoring metric. Due to differences in the shape of the histogram between cluster-like data and trajectory-like data, this entropy-based persistent homology score is expected to be smaller for cluster-like datasets and larger for trajectory-like datasets.

### Scoring metric 3 –vector magnitude

The third scoring metric is derived by calculating the magnitude of summation of a sequence of vectors in the point clouds, where the sequence of vectors is generated by connecting neighboring clusters defined by over-clustering of the point cloud. In trajectory-like data, the vectors

tend to be correlated with the general trend of the trajectory, leading to a large magnitude for the sum of the vectors. In cluster-like data, the directions of the vectors tend to be random, and the sum of the vectors tends to be small.

Given a dataset, if its dimension is greater than 5, we apply PCA to reduce the dataset's dimensionality to 5. The data is normalized so that elements in each dimension range from 0 to 1. After that, we run K-means clustering on the dataset and identify the cluster centers. The default value of K is 5% of the number of data points in the dataset. Among the cluster centers, one cluster center $A$ is chosen, and its closest (Euclidean distance) cluster center $B$ is identified, and the vector pointing from $A$ to $B$ is calculated. Now from $B$, a new cluster center $C$ with the closest distance is identified, and another vector pointing from $B$ to $C$ is calculated. The above steps are repeated until the distance to the next closest cluster center is above a distance threshold (i.e. 20th percentile of all pairwise distance among the K-means cluster centers). Once the iteration stops, all the vectors calculated are added to form an overall vector. The magnitude of the overall vector's norm is calculated using p-norm, $(\sum_{i=1}^{n} |x_i|^p)^{\frac{1}{p}}$, where $p$ is the dimesons of the PCA space, which is 5 for our default setting. Because of the stochastic nature of K-means, the above process is repeated 5 times and the average of the five magnitudes is used as the final score. We expect that trajectory-like datasets would have a greater magnitude for the overall vector because trajectory-like datasets often have strong directionality present in how data points are continuously spreading out in the space, whereas cluster-like datasets would have a relatively smaller magnitude for the overall vector.

## Scoring method 4 –Ripley's k function

The fourth score is motivated by Ripley's $k$ function. Ripley's $k$ function is a type of spatial cluster analysis tool often used for analyzing datasets with spatial interpretations [25]. Its usage is mostly limited to spatial analysis of two to three-dimensional data, such as spatial patterns of trees in a forest or locations of nests of birds. The formula for Ripley's $k$ function is as follows: $k(t) = \frac{1}{\lambda} \sum_{i \neq j} \frac{I(d_{ij} < t)}{n}$, where t refers to distance threshold, $\lambda$ refers to the average density of points, $d_{ij}$ refers to the distance between two data points $i$ and $j$, $n$ refers to the number of data points, and $I$ refer to the indicator function. The value of $k$ is measured across different values of distance threshold $t$, and the observed $k$ values for a query dataset are contrasted with $k$ values computed from uniformly distributed random spatial distributions. Overall, the Ripley's $k$ function quantifies the contrast between the data and the uniform distribution. Based on our observations, the Ripley's $k$ contrast between cluster-like data and a random uniform distribution is often larger than the contrast between trajectory-like data and the random uniform distribution. Therefore, the Ripley's $k$ function could be used to distinguish between cluster-like data and trajectory-like data.

We modified the conventional Ripley's $k$ to make it more suitable for higher-dimensional data like single-cell data. Given a dataset, we created a random dataset that is uniformly distributed within the convex space that contains all data points in the original dataset. We generate geodesic DPT distance matrices for both original and random datasets. Respective maximum values of the geodesic distances across the two distance matrices are identified, and we create 100 equally spaced distance thresholds ranging from 0 to each maximum value, serving as $t$ in the above Ripley's $k$ equation. For each distance threshold $t$, $k$ values are evaluated for the original and random datasets separately. Then, the two sets of $k$ values are normalized to between 0 and 1, the absolute differences between the two sets of normalized $k$ values are computed in an element-wise fashion, and the area under the curve of the difference is calculated. This area under the curve is our fourth scoring metric. Since Ripley's $k$ aims to capture

the patterns of clustering, we expect that this scoring metric would be able to give clustering-like datasets high scores, and give trajectory-like datasets lower scores.

## Scoring metric 5 –degrees of connectivity

The fifth scoring metric is motivated by the concept of degrees of connectivity which is often used to quantify social connections. Based on the idea of a chain of 'friend of a friend' would connect everyone, we attempted to quantify the percentage of data points connected as we vary the number of neighbors ('friends'). In a dataset that contains distinct clusters, the connections are limited within the clusters when the number of neighbors is small. In contrast, in a dataset that represents a continuous trajectory, even a small number of neighbors may enable all data points to be connected directly or indirectly. Therefore, a scoring metric quantifying connectivity has the potential to distinguish cluster-like data and trajectory-like data.

For this scoring metric, first, we generate a range of different numbers of neighbors from 5% to 95% of the number of data points. For each neighbor size as k, for each data point, we find its k nearest neighbors. Using the identified neighbors for all cells, the symmetric mutual nearest neighbor graph is constructed. If there are $n$ cells in the dataset, this graph will be an $n$-$by$-$n$ symmetric matrix with 0 or 1 entries, where an entry is 1 only if the two corresponding cells are mutually nearest neighbors of each other, i.e., each cell in a pair of cells is contained in each other cell's nearest neighbors. Using the symmetric nearest neighbor graph, we could evaluate, for each cell, how many other cells in the dataset it could reach based on not only its direct neighbors but also subsequent neighbors' neighbors, and so on. Hence, for each cell, the proportion of cells it could reach quantifies its reachability value, and the median reachability value for all data points for a specific neighbor size is calculated. The above is repeated for different neighbor sizes to generate a series of median reachability values, one for each neighbor size. The series of reachability values are then summarized into a curve where the x-axis is the proportion of neighbors, and the y-axis is the reachability values calculated. Finally, we compute the area under the curve, which is our fifth scoring metric. We expect that cluster-like datasets have relatively weak connections among the clusters, leading to low scores because each cell's reachability would be limited to its own cluster until the number of neighbors increases to be larger than its cluster's size. In contrast, in trajectory-like datasets, cells could be potentially connected with many other cells even with relatively smaller neighbor sizes, leading to higher scores.

## Simulated clustering-like and trajectory-like datasets

We generated a variety of simulated datasets to test the five scoring metrics. Specifically, we designed four types of simulated datasets, including (1) clear clusters data, (2) clear trajectory data, (3) noisy clusters data, and (4) noisy trajectory data. The simulated datasets were all two-dimensional. The clear clusters data were generated by random sampling from mixture Gaussian models, with random proportions for the components and relatively small variances for each component compared to mean differences, so that the simulated data were clearly cluster-like showing distinctive clusters with clear separations that can be visually confirmed. The clear trajectory data were generated by random sampling from sine functions to create trajectories, followed by adding a relatively small amount of Gaussian noise to create width along the trajectories, so that the simulated data were trajectory-like showing a clear progression trend like a belt. The noisy clusters datasets and noisy trajectory datasets were generated in the same way as the simulated clear clusters and clear trajectory data, except that the variance of the added noise was larger so that the cluster-like or trajectory-like patterns in the simulated dataset were less clear. For the noisy trajectory datasets, we included bifurcation cases where a

trajectory was divided into two separate branches. A total of 12,000 datasets were generated in this study, 3000 for each of the four types of simulated datasets. Each simulated dataset was scored using the fiving scoring metrics. A few examples of simulated datasets are shown in Fig 3B.

### Single-cell RNA-seq Data

scRNA-seq datasets used in this study are as follows: planaria dataset [29], mouse bone-marrow dataset [30], mouse epidermis dataset [31], monkey epiblast dataset [32], fibroblast dataset [33], natural killer T cell dataset [34], mouse cell atlas dataset [30], Tabular Sapiens dataset [35], as well as the single-cell datasets from benchmark papers for single-cell trajectory inference and clustering methods [12,13,36]. These datasets were used to evaluate robustness and accuracy of our scores in quantifying the clusterness and trajectoriness. The quality control and data pre-processing methods used in the benchmark study [13] were applied, including filtering of genes with too few non-zeros entries, filtering of cells with abnormal library size or high mitochondrial expression, library-size normalization, and highly variable gene selection. The dimensionality of the data was further reduced to 20 using PCA transformation for all scRNA-seq datasets.

### UMAP projection of the scores

For each of the 12,000 simulated datasets, we computed the five scoring metrics, resulting in a 12,000 by 5 matrix of scores. Uniform Manifold Approximation and Projection (UMAP) [37] was applied to the matrix of scores to generate a 2-dimensional visualization of the landscape of the simulated datasets in the 5-dimensional space of the scoring metrics, resulting in a dimension-reduced matrix of size 12,000 by 2. The distance metric used for UMAP was the Euclidean distance, with the number of neighbors set to be 30 and the minimum distance set to be 0.6. For a given real scRNA-seq dataset, we computed the five scoring metrics and project the resulting scores onto the UMAP of the simulated datasets. We hypothesize that simulated datasets located close to the projection of the real dataset could share similar geometric characteristics (cluster-like or trajectory-like), thereby, providing an indicator for the geometry of the real dataset.

## Results

### Proposed scores can differentiate between cluster-like and trajectory-like datasets

The simulated datasets were used to test whether the proposed scores could quantify clusterness and trajectoriness, and thus, differentiate datasets with the different underlying geometry. Violin plots for the scores across the four simulated types of data were shown in Fig 4, demonstrating that each of the five metrics had some ability to distinguish the simulated cluster-like and trajectory-like datasets. Combination of these scores could provide enhanced power to classify whether a dataset is cluster-like or trajectory-like.

### Proposed scores define a geometric landscape of clusterness and trajectoriness

Scores of the 12,000 simulated datasets across the four simulated types (clear clusters, clear trajectory, noisy clusters, and noisy trajectory) were projected to two-dimensional space using UMAP, as shown in Fig 5. In the UMAP visualization in Fig 5A, scatter plots of the simulated datasets are used to represent dots in the UMAP. Clear clusters data (bottom-right region of UMAP) and clear trajectory data (bottom-left region of UMAP) were well separated, indicating that the combination of the five scores were able to differentiate them. Noisy clusters data and noisy trajectory data were located in between the clear clusters data and clear trajectory

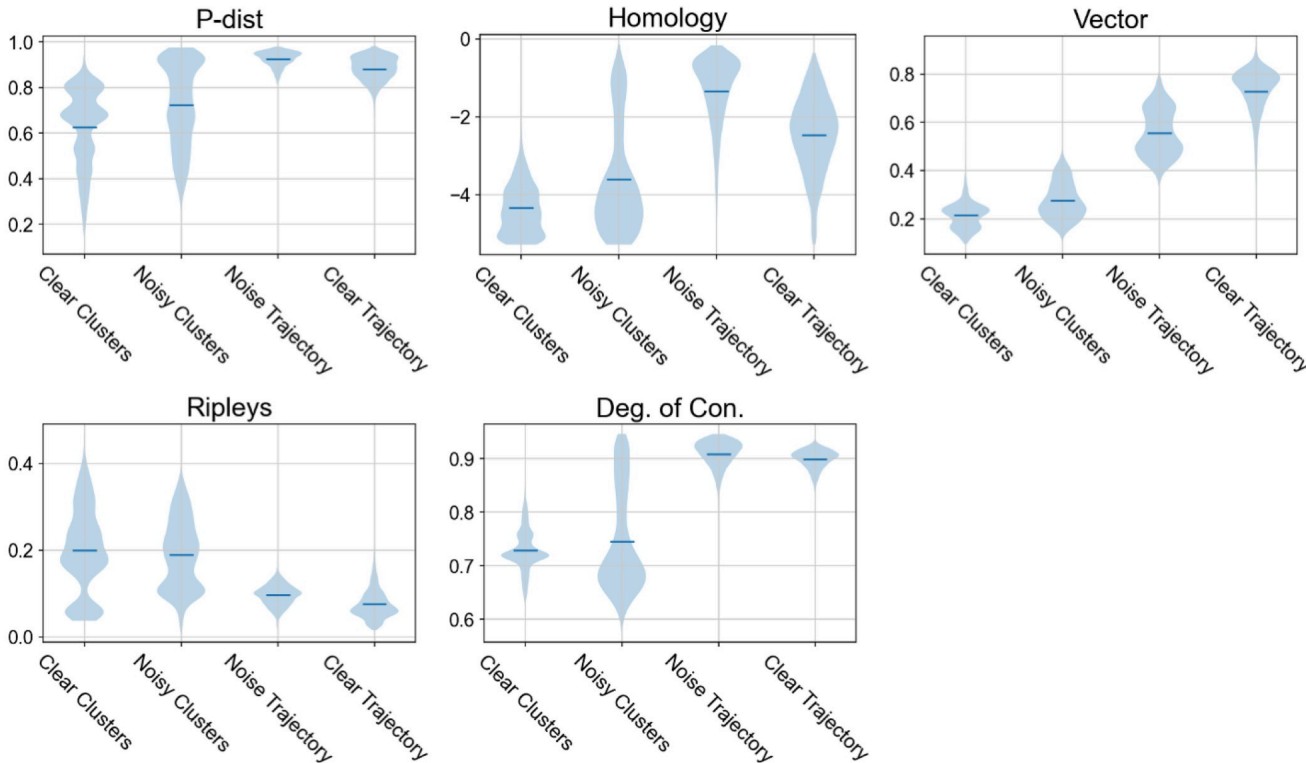

**Fig 4. Proposed scores show meaningful differences between cluster-like and trajectory-like datasets.** For each scoring metric, a violin plot shows the scores across simulated clear clusters data (n = 3000), simulated clear trajectory data (n = 3000), simulated noisy clusters data (n = 3000), and simulated noisy trajectory data (n = 3000). Blue solid lines represent the median of the distributions. All score metrics exhibit meaningful differences across the four simulated types of data.

data. In Fig 5B, dots in the UMAP plot were colored by the proportion of neighboring dots/ datasets belonging to clear or noisy trajectory-like datasets. The boundary on the colored UMAP showed the separation between noisy clusters data and noisy trajectory data. Overall, we observed that datasets with similar geometry (similar-looking scatter plots) were located close to each other on the UMAP. In addition, we observed gradual changes in clusterness and trajectoriness geometry on the UMAP in Fig 5A. The gradual changes in geometric characteristics in the UMAP space were further reflected in Fig 5C, which showed the variations of each scoring metric in the UMAP space. In addition to the dichotomy of clusterness and trajectoriness, Fig 5C also showed variation of some metrics within one type of geometry. Within the right side of cluster-like data, we observed variations in P-dist, Ripley's k and Degree of Connectivity, especially at the bottom-right tip. This was because the bottom right tip corresponded to cluster-like data with larger number of clusters compared to the rest of cluster-like datasets. Within the left side of trajectory-like data, the Homology metric showed a gradient that correlated to the level of noise we simulated, where smaller amount of simulation noise (i.e., narrower width of the trajectories) led to smaller value of the Homology metric.

## Mapping scRNA-seq datasets to the clusterness and trajectoriness geometric landscape

To evaluate whether the simulated clusterness and trajectoriness geometric landscape is generally applicable, we collected 169 scRNA-seq datasets from previously published benchmark

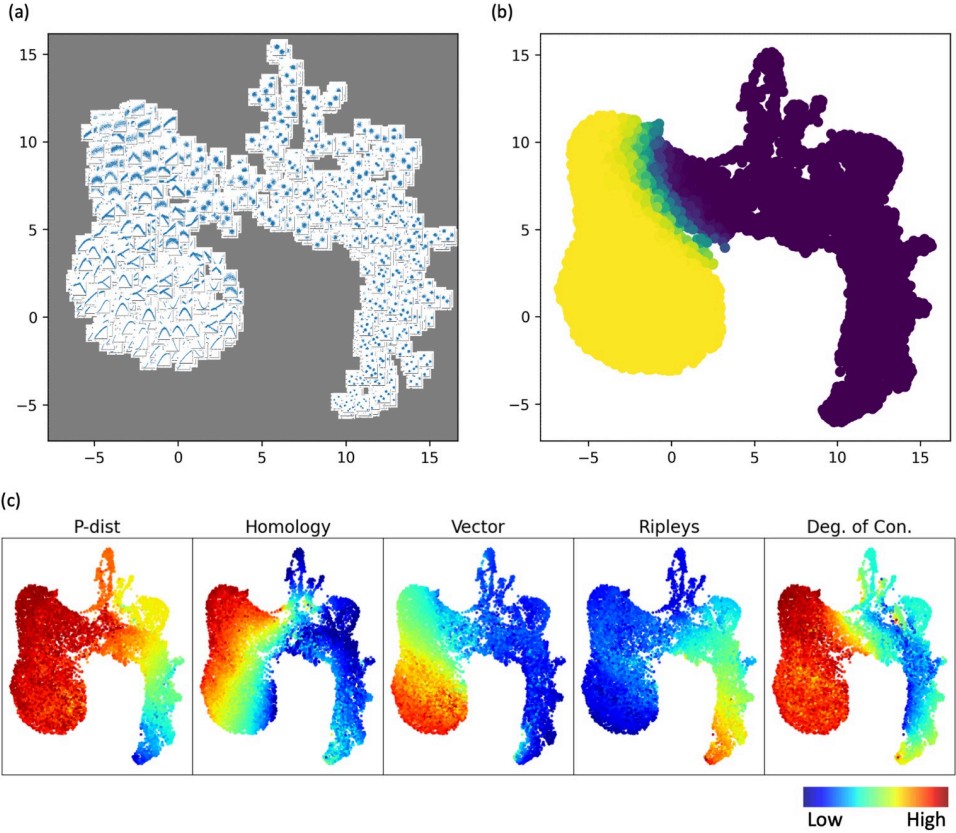

**Fig 5. Simulated geometric landscape of clusterness and trajectoriness.** (a) Each dot in the UMAP represents one simulated dataset, and is visualized by the scatter plot of the simulated dataset itself. (b) Dots in the UMAP plot were colored by the proportion of neighboring dots belonging clear or noisy trajectory-like data. The boundary on the colored UMAP showed the separation between cluster-like and trajectory-like datasets. (c) Colored visualizations by values of each of the five scoring metrics, showing variations of the scores in the UMAP space. Red represents higher values, and blue represents lower values.

studies for clustering and trajectory inference algorithms, computed the scoring metrics for these datasets, and projected the resulting scores onto the clusterness and trajectoriness geometric landscape defined by the simulated datasets. As shown in Fig 6A, the projections of these 169 real datasets scattered and covered both cluster-like and trajectory-like regions of the geometric landscape. The overlap between the projections of real and simulated datasets indicated that scores were comparable. Fig 6B further demonstrated that the range and distribution of the scores were indeed comparable between simulated and real datasets.

These 169 datasets were considered by previous studies as suitable benchmarking data for either clustering analysis or trajectory inference because of researchers' presumed geometric intuitions based on the experimental design and underlying biology. The presumed geometry of these datasets included "clusters", "organs", "disconnected graph", which should be cluster-like. The presumed geometry also included "tree", "convergence", "linear", "cycle", "acyclic graph", "bifurcation", "multifurcation", which should be trajectory-like. After projecting these 169 datasets onto our clusterness and trajectoriness landscape, we applied the k-nearest neighbors idea to classify each dataset based on the location of its projection, and thereby, generating our computational prediction of whether the datasets were cluster-like or trajectory-like. The table in Fig 6C summarized the prediction results organized by the presumed geometry types. For "clusters", "organs" and "disconnected graph" datasets which were presumed to be

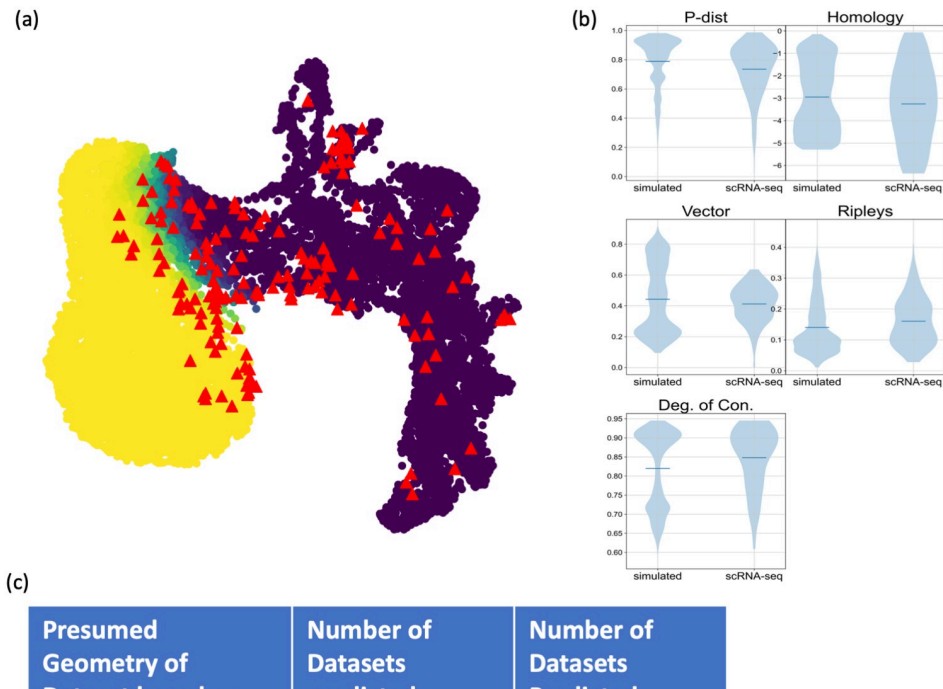

(a)

(b)

(c)

| Presumed Geometry of Dataset based on Biology | Number of Datasets predicted Cluster-like | Number of Datasets Predicted Trajectory-like |
|---|---|---|
| Cluster | 19 | 13 |
| Organs | 21 | 7 |
| Disconnected Graph | 23 | 2 |
| Tree | 17 | 2 |
| Convergence | 1 | 0 |
| Linear | 17 | 22 |
| Cycle | 1 | 1 |
| Acyclic Graph | 1 | 0 |
| Bifurcation | 7 | 6 |
| Multifurcation | 7 | 2 |

**Fig 6. Projections of scRNA-seq datasets onto the simulated geometric landscape.** (a) Each red triangle represents the projection of one of the 169 real scRNA-seq datasets projected to the simulated geometric landscape. (b) Violin plots of the scoring metrics in the simulated data versus the real scRNA-seq data, showing that the distribution of scores were similar between simulated and real datasets. (c) A tabular summary of the presumed geometric intuition and predicted geometric property for the 169 datasets, showing roughly 70% agreement.

cluster-like, 74% of them were predicted to be cluster-like in our approach. For "linear" which was presumed to be the most trajectory-like, 56% were predicted to be trajectory-like in our approach. For the "tree", "bifurcation" and "multifurcation" datasets, the presumed geometry and our prediction were uncorrelated and even somewhat opposite. The number of datasets in other types were too few to meaningfully compare. Overall, this comparison showed that the presumed geometry based on biological intuition and the prediction based on our quantitative

metrics agreed in roughly 70% of the datasets. This agreement was an encouraging validation of the proposed clusterness and trajectoriness geometric landscape. In the meantime, the disagreement was alarming, indicating that the presumed geometry based on biological intuition may be questionable in up to 30% of the cases.

## Clustering reduces both clusterness and trajectoriness of the data

For each of the 169 scRNA-seq data above, we applied the Seurat package to cluster the data, and divided one dataset into multiple datasets based on the clustering result, so that we obtained smaller datasets that correspond to individual clusters derived from the 169 datasets. We then scored these smaller datasets and mapped them to the clusterness and trajectoriness geometric landscape, shown as the red dots in Fig 7A. Compared to the projections of the 169 datasets shown in Fig 6A, projections of these individual clusters were more enriched toward the boundary region in the geometric landscape. Fig 7B highlighted one cluster-like dataset whose projection was the red triangle deep in the cluster-like region of the geometric landscape, while projections of its clusters were all in the region occupied by simulated noisy trajectory-like datasets. This example showed that individual clusters of a cluster-like datasets were less cluster-like, which was expected. Fig 7C highlighted a trajectory-like dataset whose projection was in the trajectory-like region of the geometric landscape, whereas projections of its

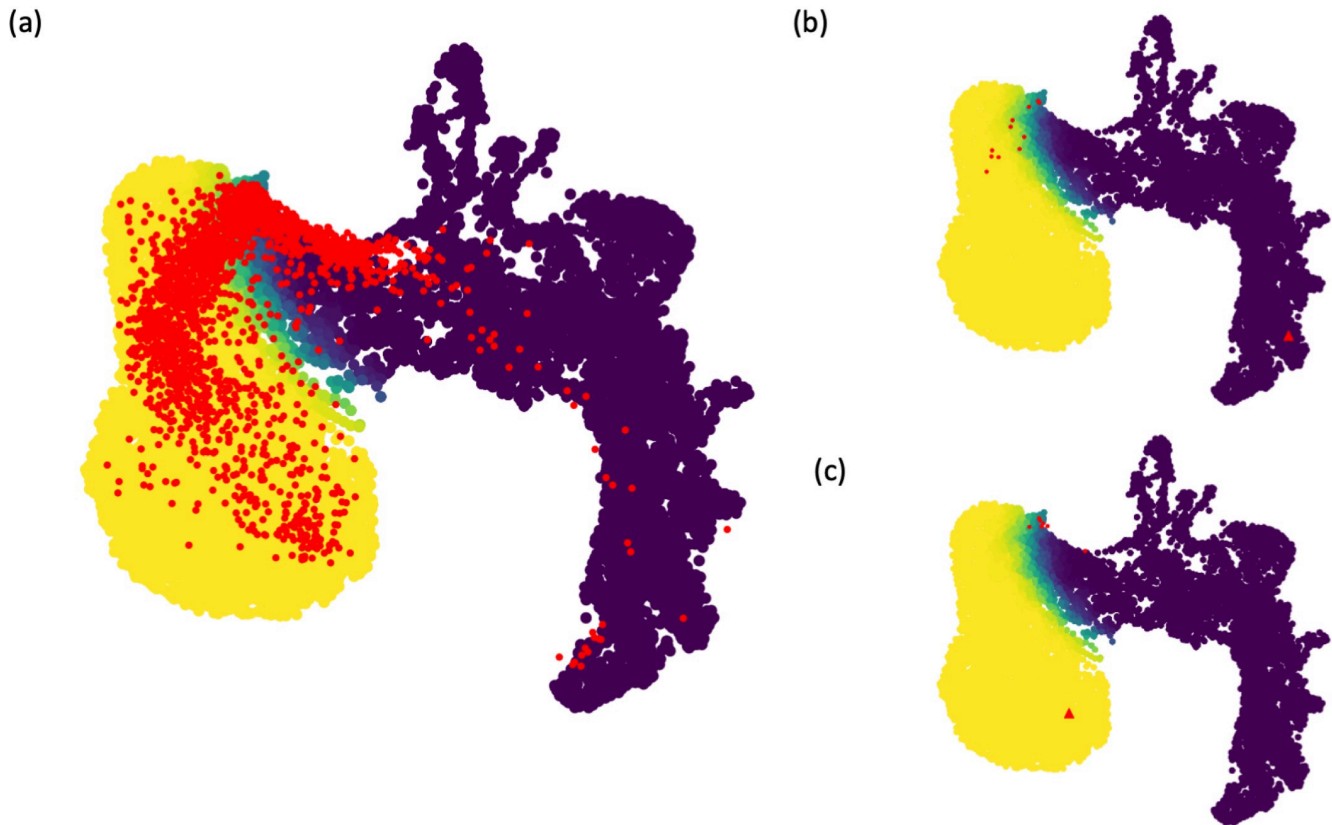

**Fig 7. Projection of individual clusters onto the simulated geometric landscape.** (a) After clustering analysis of the 169 scRNA-seq datasets, each resulting cluster is treated as a separate dataset and projected onto the simulated geometric landscape. The resulting projections, shown as the red dots were enriched toward the boundary between cluster-like and trajectory-like regions of the geometric landscape. (b) Projections of one cluster-like dataset and its individual clusters, shown by the red triangle and red dots respectively. (c) Projections of one trajectory-like dataset and its individual clusters, shown by the red triangle and red dots respectively.

clusters were again in the region occupied by simulated noisy trajectory-like datasets. This was also expected, because clustering would divide a long trajectory into shorter pieces whose trajectoriness became less prominent.

## Geometric landscape as a guide for choosing proper analysis for interpretation

We examined datasets that showed clear trajectory-like geometry in previous studies. Two published datasets on mouse placenta trophoblast and inter-follicular epidermis [30,31,38] were analyzed using both clustering and trajectory-inference methods. As shown in the two panels in Fig 1A and 1B, both types of analyses suggested these two datasets were trajectory-like, which was consistent with the fact that these two datasets were generated for understanding continuum of progressions, defined by the development to cell types in the placenta trophoblast [39] and the differentiation from the epidermis stem cells to various epidermis cell types [40]. We computed the proposed scoring metrics for these two datasets, and projected the scores to the simulated geometric landscape. In Fig 8A, we observed that these two datasets were both projected to the left-side of the UMAP where simulated trajectory-like datasets were located, meaning that the proposed scoring metrics also considered these two datasets to be trajectory-like. Similarly, we examined two datasets that showed clear cluster-like geometry. Two published datasets on the mouse cell atlas and the monkey epiblast [29,32] were analyzed using both clustering and trajectory-inference methods, and the results indicated clearly distinctive clusters as visualized in the two panels of Fig 1C and 1D. The cluster-like geometry was expected, because of the diversity of the mouse cell atlas covering multiple distinct organs [41] and the diverging pluripotent nature of epiblast [42]. We computed the proposed scoring metrics for these two datasets, and their projections to the simulated geometric landscape were shown in Fig 8A. We observed that these two datasets were projected to the middle and the right side of the UMAP where simulated cluster-like datasets were located, meaning that the proposed scoring metrics correctly captured the cluster-like nature of these two datasets.

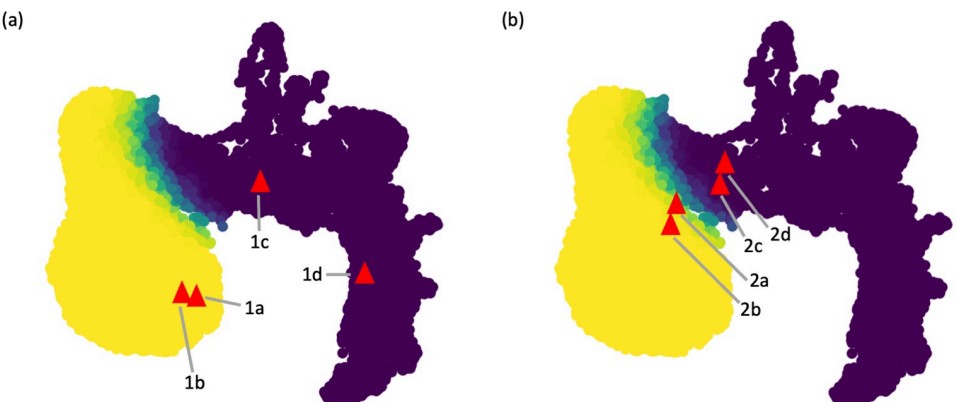

**Fig 8. Projections of example scRNA-seq datasets to the simulated geometric landscape.** (a) Projections of the four datasets visualized in Fig 1 to the simulated geometric landscape. The two datasets with clear trajectory-like visualizations in Fig 1A and 1B are both mapped to the bottom-left side of the UMAP landscape which contains predominantly simulated trajectory-like datasets. The two datasets with clear clustering-like visualizations in Fig 1C and 1D are both mapped to the mid and right side of the UMAP landscape which contains mostly simulated cluster-like datasets. (b) Projection of the four datasets visualized in Fig 2, whose geometric interpretations are different based on clustering and trajectory inference analyses. The two datasets in Fig 2A and 2B are mapped to the left side of the geometric landscape, surrounded by simulated trajectory-like datasets. The two datasets in Fig 2C and 2D are mapped to the middle region of the geometric landscape, surrounded by simulated cluster-like datasets.

We further examined four scRNA-seq datasets whose geometry was less clear based on existing analysis methods. These four datasets were generated in the contexts of bone marrow mesenchyme differentiation, mouse embryonic fibroblast, planaria whole-body, and natural killer cell subtypes [29, 30, 33, 34]. We analyzed these four datasets using both clustering and trajectory-inference methods, and visualized the results in Fig 2. First panel of Fig 2A–2D visualized results of the clustering analysis, and showed distinctive cell clusters underlying these four datasets. In contrast, second panel of Fig 2A–2D provided visualizations of trajectory-inference analysis, which revealed trajectories underlying these four datasets. Given the contradictory geometry from clustering and trajectory inference of these datasets, it was difficult to conclude whether these datasets were cluster-like or trajectory-like. We computed our scoring metrics for these four datasets, and projected them to the simulated geometric landscape as shown in Fig 8B. The first two datasets were mapped to the left side of the geometric landscape, surrounded mostly by trajectory-like simulated datasets, indicating that these two datasets were both trajectory-like. Since these two datasets were generated to study mouse bone marrow mesenchyme erythrocyte differentiation [43] and the induction of fibroblast to neuronal cells [33], it is plausible that these datasets contain trajectories driven by the differentiation processes. The third and fourth datasets were mapped to the middle region of the geometric landscape which contained mostly cluster-like simulated datasets, suggesting that these two datasets have cluster-like geometry containing distinct cell types, such as the gut, muscle, neuronal, epidermal cells in the planaria differentiated from neoblasts [29], as well as various subtypes of natural killer T cells [34]. From these four examples, we observed that the projected location of a scRNA-seq dataset on the simulated geometric landscape could capture the clusterness and trajectoriness of the data, which facilitates our understanding of the distribution of the data and provides guidance of whether clustering analysis or trajectory inference would be more appropriate for interpreting the data.

## Availability and future directions

In the literature of computational analysis of scRNA-seq, a common practice is to choose analysis approach based on the biological intuition and expected geometry of the experimental design that generates the data. However, we demonstrated example scRNA-seq datasets where clustering and trajectory inference produced drastically different visualizations and interpretations of the same data. Since clustering analysis tends to produce clusters and trajectory inference tends to generate trajectories, selecting analysis approach based on the expected geometry can introduce bias that reinforces the prior expectations. This is potentially dangerous and misleading in cases where the actual geometry of the data is different from expected. Therefore, it is important to be aware of such bias, and develop objective methods to quantify the geometry of scRNA-seq data, which could provide a guide for selecting appropriate analysis approach for biological interpretation.

We present five scoring metrics and a computational pipeline to quantify the clusterness and trajectoriness of scRNA-seq data. The proposed scoring metrics are based on pairwise distance distribution, persistent homology, vector magnitude, Ripley's K, and degrees of connectivity. Using simulated datasets, we demonstrated that these scoring metrics are able to differentiate cluster-like data and trajectory-like data. UMAP visualization of the scores from simulated data revealed a geometric landscape of clusterness vs. trajectoriness. With a large collection of real scRNA-seq datasets, we showed that the proposed scoring metrics and the simulated geometric landscape are applicable to real scRNA-seq data. We demonstrated the utility of the simulated geometric landscape to infer the geometric characteristics of real scRNA-seq data, which could serve as an indicator for choosing whether clustering or trajectory inference is a more appropriate type of analysis approach for interpreting a given dataset.

We acknowledge that there may exist alternative metrics for geometric quantification of scRNA-seq data. The five metrics presented here were tailored to capture intuitive differences between the two types of geometry, and each of them indeed exhibited a decent level of discriminatory power. An interesting future direction is to develop additional metrics that can lead to insights beyond the five presented here, and examine whether any additional metric is able to further enrich the variations captured by the geometric landscape.

Previous benchmarking studies of clustering and trajectory inference algorithms provided extensive datasets with presumed geometric intuitions of whether those datasets were suitable for clustering or trajectory inference. Those benchmarking datasets were valuable resources for validating the proposed clusterness and trajectoriness geometric landscape. When projecting those scRNA-seq datasets onto the clusterness and trajectoriness geometric landscape, the predicted geometry and the presumed geometry agreed in roughly 70% of the datasets. This result was both encouraging and alarming, especially because the 30% disagreements were observed among datasets that were previously chosen for benchmarking analyses of clustering and trajectory inference. This result suggested a non-trivial possibility that how an scRNA-seq dataset looks like is different from how we think it should look like. Beyond the specific scoring metrics and geometric landscape, a more general goal of this study is to inspire discussions about quantifying data geometry and choosing appropriate analysis type in an objective and quantitative manner, which is an important question that has not been recognized and discussed in the single-cell research community.

The concept of clusterness and trajectoriness may not be sufficient to fully capture the rich geometry that scRNA-seq data can exhibit. There definitely can exist datasets with both cluster-like and trajectory-like characteristics. One example is "clusters of trajectories" where a dataset could be composed of distinct clusters and each cluster itself is a trajectory. Another example is "a trajectory of clusters" where a dataset could be made of spherical-shaped clusters that are separated but lined up to form a trajectory. When projecting such datasets onto the clusterness and trajectoriness geometric landscape, the predicted geometry may not be robust and will be dependent on the prominence of the trajectories and separation among the clusters in the data. As a potential solution, we could first apply clustering analysis to the data, and then evaluate the clusterness and trajectoriness for both the original dataset and the individual clusters. The predicted geometry of the original dataset and its clusters collectively can provide more detailed descriptions of the geometric characteristics of the original dataset. Therefore, quantification of clusterness and trajectoriness can serve as a good first attempt toward developing a comprehensive set of descriptors for characterizing the geometry of scRNA-seq data.

The code for the proposed geometric quantification, along with use examples and documentations, is available at https://github.com/pqiu/Quantifying_clusterness_trajectoriness

## Author Contributions

**Conceptualization:** Peng Qiu.

**Data curation:** Hong Seo Lim.

**Formal analysis:** Hong Seo Lim, Peng Qiu.

**Funding acquisition:** Peng Qiu.

**Investigation:** Hong Seo Lim, Peng Qiu.

**Methodology:** Hong Seo Lim, Peng Qiu.

**Project administration:** Peng Qiu.

**Resources:** Peng Qiu.

**Software:** Hong Seo Lim.

**Supervision:** Peng Qiu.

**Validation:** Hong Seo Lim, Peng Qiu.

**Visualization:** Hong Seo Lim, Peng Qiu.

**Writing – original draft:** Hong Seo Lim, Peng Qiu.

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
