## [Decision Letter · Decision Letter 0]

8 Oct 2023

Dear Dr. Qiu,

Thank you very much for submitting your manuscript "Quantifying the clusterness and trajectoriness of single-cell RNA-seq data" for consideration at PLOS Computational Biology.

As with all papers reviewed by the journal, your manuscript was reviewed by members of the editorial board and by several independent reviewers. In light of the reviews (below this email), we would like to invite the resubmission of a significantly-revised version that takes into account the reviewers' comments.

We cannot make any decision about publication until we have seen the revised manuscript and your response to the reviewers' comments. Your revised manuscript is also likely to be sent to reviewers for further evaluation.

Sincerely,

Shihua Zhang

Academic Editor

PLOS Computational Biology

Jian Ma

Section Editor

PLOS Computational Biology

Reviewer's Responses to Questions

**Comments to the Authors:**

Reviewer #1: The authors of this submission has presented to quantify "clusterness" and "trajectoriness" of scRNA-seq data, which could help researchers to decide what would be suitable analysis methods given scRNA-seq data.

This review applauds the efforts of developing such a quantitative computational pipeline and believes that the presented work can indeed help researchers, especially biomedical researchers, to better explore and understand data.

There are a few questions/concerns that the authors may want to address:

1. Are the selected five metrics, pairwise distance distribution, persistent homology, vector magnitude, Ripley's K, and degrees of separation, comprehensive? Are there other possibly better metrics to use? The authors may want to discuss on the motivations of using these five specific metrics. Also, as computing some of these metrics depends on some selected algorithms or hyperparameters, will the final "clusterness" and "trajectoriness" results change depending on these choices? Finally, it appears that higher degree of separation scores indicates better connectivity (or worse separation). Is this historically this way or it is better to call it degree of connectivity instead of separation?

2. The generative 'geometric landscape' based on simulated data is using UMAP. Will the results depend on different embedding / projection methods?

3. Such geometric landscapes can probably change with simulated data, for example, the smoothness of trajectories, the number of clusters, etc., which do not necessarily help derive consistent/useful "clusterness" and "trajectoriness" scores? How would the authors take care of these?

4. It is indeed encouraging to see the results in Figs 4 and 5. However, the authors may want to discuss some non-intuitive situations. For example, why some single clusters still appear in the right 'clusterness' branch in 'geometric landscape' in Fig 5a? In Fig 4b, it might be better to separately show the distributions of these metrics for "cluster"-like and "trajectory"-like scRNA-seq data to indeed check whether they are similar as simulated data, instead of pulling all 169 datasets together? Regarding the biological ground truth, are these labels in Fig 4c based on 20 PCs as described in the text or there are more biologically grounded knowledge?

5. In the generated landscape in Fig. 3, it is non-intuitive that both P-dist and degree of separation scores are high at the right-bottom tip. The authors may want to provide more discussions to help better understand the potential reason/limitation?

Reviewer #2: Summary of Article

This article aims to address a key challenge in scRNA sequencing analyses. They discuss two key approaches to scRNA seq analysis: cluster analysis and trajectory inference analysis. Each of these approaches operate under distinct assumptions. This article aims to quantify the strength of each approach based on the geometric characteristics of the data. The authors present scoring metrics that aim to quantify “clusterness” and “trajectoriness” of the data. The authors use simulated data sets to differentiate between cluster-like data sets and trajectory-like datasets. They apply their metrics on real scRNA seq data sets to demonsrate how the metrics can be used as indicators of whether clustering or trajectory inference is appropriate for the data. This article would be of interest to the community of researchers using scRNA seq. It offers tools that can be used to help with the interpretation of data. With some revisions, this article is suitable for publication.

Methods:

The 5 scores that the authors use are well described in the methods and absolutely essential to understanding the manuscript though there are some inconsistencies between what the authors expect and the outcomes in the figures. Comments are below in the detailed notes.

The Github repository does share the code associated with the manuscript, but the README is empty and there is very little information on how to reuse the data set or run the analysis. Without a great deal of experience, this could would be difficult or impossible to reuse. The code would benefit substantially from additional annotation on what the different steps of the code are doing, order in which files would need to be used, etc. If someone had their own data set, where would it be brought into the pipeline so they can do their own comparison?

Detailed Notes

Figure 1: Depiction of approach. No comments

Figure 2: Figure 2 depiction of score distributions grouped by set-type using simulated data sets. I agree this shows some indication of separation between clustered data sets and trajectory data sets.

Figure 3: UMAP clustering of 5-score outcomes from simulated data indicate separation between clearly clustered data sets and clearly trajectory data

3c – It appears that there is a gradient low-to-high for all metrics defined by the authors except for Homology and Degree of Separation. Can the authors comment on why these do or do not match the expected outcomes described in the methods section?

Figure 4: The authors depict where the real scRNA seq data sets fall along their UMAP spectrum of simulated data sets.

4a – Many of the data sets fall somewhere in the middle of the UMAP spectrum. It would be interesting to see a supplemental figure depicting some examples of scRNA seq data sets that fall in the middle where they are not exactly determined to be cluster like but also not exactly trajectory-like. In practice, someone might want to assess the clusteriness or trajectoriness characteristics in their data sets, but examples of how they might interpret data if it is not clearly one or the other would be very helpful for a reader.

4b – It appears as though the range of the data sets for all 5 metrics are similar between the simulated and scRNA-seq data sets, but the distributions are not all similar. The vector measurements and the degree of separation are particularly different. Can the authors comment on why the distributions are different between the scRNA seq and simulated data sets for these two metrics.

4c – The authors mention around a 70% agreement between the prediction of cluster-like/ trajectory-like geometry for real scRNA seq data sets and their presumed geometric intuitions based on the experimental design and underlying biology. It might benefit readers to see the scores and distributions of those that agreed versus the scores and distributions of those that did not agree. While the n is low for some of the presumed geometries, there could be enough data sets to display this for Cluster, Organs, Tree, Linear and possibly Bifurcation.

Figure 5: Authors clustered the scores for the 169 scRNA seq data sets using Seurat. It is not entirely clear what this figure is adding to the manuscript. It appears the authors are adding an additional transformation to the data set but clustering in Seurat and seeing if it holds up in their geometric landscape. Since the data are already abstracted through the use of the scores, adding an additional layer of clustering feels difficult to interpret. I might rather see a comparison of score distributions for data sets that agreed compared to those that did not agree as mentioned in 4c.

Figure 6: Representative examples of how real scRNA seq data sets described as trajectories and clusters fall into the geometric landscape. The examples are clear and illustrative.

Figure 7: Representative examples of how real scRNA seq data sets, that are not clearly described as trajectories or clusters, fall into the geometric landscape. The authors show that their method can help determine the clusterness and trajectoryness for data sets that are not clearly described as either.

Discussion: Overall, this manuscript is interesting. It offers tools to researchers to help with the determination of whether to use cluster analysis or trajectory inference analysis on a data set. While I agree with the authors that using one analysis will bias the outcome towards the type of analysis conducted (ie. clustering analysis will generate clusters) the authors could benefit from describing examples of where this could make a difference when studying the biological significance of different pathways. It does not feel sufficient to say that it will help us distinguish between tragectories of clusters or clusters of trajectories, why is this important biologically? It is always useful to have more tools to work with, but it remains unclear how this makes a difference in our fundamental study of biological systems.

Reviewer #3: The authors provide a report that is centered around the development of a method/pipeline to evaluate / quantifiy “clusterness” and “trajectoriness” of scRNA-seq data. To achieve this, the authors propose five scores, namely pairwise distance distribution, persistent homology, vector magnitude, Ripley's K, and degrees of separation to quantify the “clusterness” and “trajectoriness” of single-cell RNA-seq data.

The subject is current and extremely relevant, and this beautiful study brings further understanding of the choices on how to analyze and interpret single-cell RNA-Seq data.

Since many of the single cell RNA-Seq datasets show both cluster-like and trajectory-like characteristics, I suggest that the authors apply clustering analysis to data showing this feature, and then evaluate the clusterness and trajectoriness for both the original dataset and the individual clusters. This would greatly improve the original manuscript.

**Have the authors made all data and (if applicable) computational code underlying the findings in their manuscript fully available?**

Reviewer #1: Yes

Reviewer #2: Yes

Reviewer #3: Yes

PLOS authors have the option to publish the peer review history of their article (what does this mean?). If published, this will include your full peer review and any attached files.

Reviewer #1: No

Reviewer #2: No

Reviewer #3: No
---

## [Decision Letter · Decision Letter 1]

28 Jan 2024

Dear Dr. Qiu,

We are pleased to inform you that your manuscript 'Quantifying the clusterness and trajectoriness of single-cell RNA-seq data' has been provisionally accepted for publication in PLOS Computational Biology.

Best regards,

Jian Ma

Section Editor

PLOS Computational Biology

Reviewer's Responses to Questions

**Comments to the Authors:**

Reviewer #1: The revised manuscript and discussions in the response letter have addressed all my questions.

Reviewer #2: I feel satisfied with how the authors responded to my comments and the comments of my fellow reviewers and have no further comments.

Reviewer #3: The authors answered satisfactorily all the questions raised by the reviewers.

**Have the authors made all data and (if applicable) computational code underlying the findings in their manuscript fully available?**

Reviewer #1: None

Reviewer #2: Yes

Reviewer #3: Yes

PLOS authors have the option to publish the peer review history of their article (what does this mean?). If published, this will include your full peer review and any attached files.

Reviewer #1: No

Reviewer #2: **Yes: **Ibraheem Ali

Reviewer #3: No

---

## [Editor Report · Acceptance letter]

21 Feb 2024

PCOMPBIOL-D-23-01334R1 

Quantifying the clusterness and trajectoriness of single-cell RNA-seq data

Dear Dr Qiu,

I am pleased to inform you that your manuscript has been formally accepted for publication in PLOS Computational Biology. Your manuscript is now with our production department and you will be notified of the publication date in due course.

With kind regards,

Anita Estes
